# AUCTION LEARNING AS A TWO-PLAYER GAME

**Jad Rahme**[*], **Samy Jelassi**, **S. Matthew Weinberg**
Princeton University
Princeton, NJ 08540, USA
`{jrahme, sjelassi, smweinberg }@princeton.edu`

## ABSTRACT

Designing an incentive compatible auction that maximizes expected revenue is a central problem in Auction Design. While theoretical approaches to the problem have hit some limits, a recent research direction initiated by Duetting et al. (2019) consists in building neural network architectures to find optimal auctions. We propose two conceptual deviations from their approach which result in enhanced performance. First, we use recent results in theoretical auction design to introduce a *time-independent* Lagrangian. This not only circumvents the need for an expensive hyper-parameter search (as in prior work), but also provides a single metric to compare the performance of two auctions (absent from prior work). Second, the optimization procedure in previous work uses an inner maximization loop to compute optimal misreports. We amortize this process through the introduction of an additional neural network. We demonstrate the effectiveness of our approach by learning competitive or strictly improved auctions compared to prior work. Both results together further imply a novel formulation of Auction Design as a two-player game with stationary utility functions.

## 1 INTRODUCTION

Efficiently designing truthful auctions is a core problem in Mathematical Economics. Concrete examples include the sponsored search auctions conducted by companies as Google or auctions run on platforms as eBay. Following seminal work of Vickrey (Vickrey, 1961) and Myerson (Myerson, 1981), auctions are typically studied in the *independent private valuations* model: each bidder has a valuation function over items, and their payoff depends only on the items they receive. Moreover, the auctioneer knows aggregate information about the population that each bidder comes from, modeled as a distribution over valuation functions, but does not know precisely each bidder's valuation (outside of any information in this Bayesian prior). A major difficulty in designing auctions is that valuations are private and bidders need to be incentivized to report their valuations truthfully. The goal of the auctioneer is to design an incentive compatible auction which maximizes expected revenue.

Auction Design has existed as a rigorous mathematical field for several decades and yet, complete characterizations of the optimal auction only exist for a few settings. While Myerson's Nobel prize-winning work provides a clean characterization of the single-item optimum (Myerson, 1981), optimal *multi-item* auctions provably suffer from numerous formal measures of intractability (including computational intractability, high description complexity, non-monotonicity, and others) (Daskalakis et al., 2014; Chen et al., 2014; 2015; 2018; Hart & Reny, 2015; Thanassoulis, 2004).

An orthogonal line of work instead develops deep learning architectures to find the optimal auction. Duetting et al. (2019) initiated this direction by proposing RegretNet, a feed-forward architecture. They frame the auction design problem as a constrained learning problem and lift the constraints into the objective via the augmented Lagrangian method. Training RegretNet involves optimizing this Lagrangian-penalized objective, while simultaneously updating network parameters and the Lagrangian multipliers themselves. This architecture produces impressive results: recovering near-optimal auctions in several known multi-item settings, and discovering new mechanisms when a theoretical optimum is unknown.

Yet, this approach presents several limitations. On the conceptual front, our main insight is a connection to an exciting line of recent works (Hartline & Lucier, 2010; Hartline et al., 2011; Bei &

---

[*]Corresponding author

Huang, 2011; Daskalakis & Weinberg, 2012; Rubinstein & Weinberg, 2018; Dughmi et al., 2017; Cai et al., 2019) on $\varepsilon$-truthful-to-truthful reductions.[1] On the technical front, we identify three areas for improvement. First, their architecture is difficult to train in practice as the objective is non-stationary. Specifically, the Lagrangian multipliers are time-dependent and they increase following a pre-defined schedule, which requires careful hyperparameter tuning (see §3.1 for experiments illustrating this). Leveraging the aforementioned works in Auction Theory, we propose a *stationary* Lagrangian objective. Second, all prior work inevitably finds auctions which are not *precisely* incentive compatible, and does not provide a metric to compare, say, an auction with revenue 1.01 which is 0.002-truthful, or one with revenue 1 which is 0.001-truthful. We argue that our stationary Lagrangian objective serves as a good metric (and that the second auction of our short example is "better" for our metric). Finally, their training procedure requires an inner-loop optimization (essentially, this inner loop is the bidders trying to maximize utility in the current auction), which is itself computationally expensive. We use amortized optimization to make this process more efficient.

## CONTRIBUTIONS

This paper leverages recent work in Auction Theory to formulate the learning of revenue-optimal auctions as a two-player game. We develop a new algorithm ALGnet (Auction Learning Game network) that produces competitive or better results compared to Duetting et al. (2019)'s RegretNet. In addition to the conceptual contributions, our approach yields the following improvements (as RegretNet is already learning near-optimal auctions, our improvement over RegretNet is not due to significantly higher optimal revenues).

- *Easier hyper-parameter tuning*: By constructing a time-independent loss function, we circumvent the need to search for an adequate parameter scheduling. Our formulation also involves less hyperparameters, which makes it more robust.
- *A metric to compare auctions*: We propose a metric to compare the quality of two auctions which are not incentive compatible.
- *More efficient training*: We replace the inner-loop optimization of prior work with a neural network, which makes training more efficient.
- *Online auctions*: Since the learning formulation is time-invariant, ALGnet is able to quickly adapt in auctions where the bidders' valuation distributions varies over time. Such setting appears for instance in the online posted pricing problem studied in Bubeck et al. (2017).

Furthermore, these technical contributions together now imply a novel formulation of auction learning as a two-player game (not zero-sum) between an auctioneer and a misreporter. The auctioneer is trying to design an incentive compatible auction that maximizes revenue while the misreporter is trying to identify breaches in the truthfulness of these auctions. The paper decomposes as follows. Section 2 introduces the standard notions of auction design. Section 3 presents our game formulation for auction learning. Section 4 provides a description of ALGnet and its training procedure. Finally, Section 5 presents numerical evidence for the effectiveness of our approach.

## RELATED WORK

**Auction design and machine learning.** Machine learning and computational learning theory have been used in several ways to design auctions from samples of bidder valuations. Machine learning has been used to analyze the sample complexity of designing optimal revenue-maximizing auctions. This includes the framework of single-parameter settings (Morgenstern & Roughgarden, 2015; Huang et al., 2018; Hartline & Taggart, 2019; Roughgarden & Schrijvers, 2016; Gonczarowski & Nisan, 2017; Guo et al., 2019), multi-item auctions (Dughmi et al., 2014; Gonczarowski & Weinberg, 2018), combinatorial auctions (Balcan et al., 2016; Morgenstern & Roughgarden, 2016; Syrgkanis, 2017) and allocation mechanisms (Narasimhan & Parkes, 2016). Other works have leveraged machine learning to optimize different aspects of mechanisms (Lahaie, 2011; Dütting et al., 2015). Our approach is different as we build a deep learning architecture for auction design.

**Auction design and deep learning.** While Duetting et al. (2019) is the first paper to design auctions through deep learning, several other paper followed-up this work. Feng et al. (2018) extended it to budget constrained bidders, Golowich et al. (2018) to the facility location problem. Tacchetti et al. (2019) built architectures based on the Vickrey- Clarke-Groves mechanism. Rahme et al. (2021) used permutation-equivariant networks to design symmetric auctions. Shen et al. (2019) and Duetting

---

[1] By $\varepsilon$-truthful, we mean the expected total regret $R$ is bounded by $\varepsilon$. See Prop. 1 for a definition of $R$.

et al. (2019) proposed architectures that *exactly* satisfy incentive compatibility but are specific to *single-bidder* settings. While all the previously mentioned papers consider a non-stationary objective function, we formulate a time-invariant objective that is easier to train and that makes comparisons between mechanisms possible.

## 2 Auction design as a time-varying learning problem

We first review the framework of auction design and the problem of finding truthful mechanisms. We then recall the learning problem proposed by Duetting et al. (2019) to find optimal auctions.

### 2.1 Auction design and linear program

**Auction design.** We consider an auction with $n$ bidders and $m$ items. We will denote by $N = \{1, \ldots, n\}$ and $M = \{1, \ldots, m\}$ the set of bidders and items. Each bidder $i$ values item $j$ at a valuation denoted $v_{ij}$. We will focus on *additive* auctions. These are auctions where the value of a set $S$ of items is equal to the sum of the values of the elements in that set at $\sum_{j \in S} v_{ij}$. Additive auctions are perhaps the most well-studied setting in multi-item auction design (Hart & Nisan, 2012; Li & Yao, 2013; Daskalakis et al., 2014; Cai et al., 2016; Daskalakis et al., 2017).

The auctioneer does not know the exact valuation profile $V = (v_{ij})_{i \in N, j \in M}$ of the bidders in advance but he does know the distribution from which they are drawn: the valuation vector of bidder $i$, $\vec{v}_i = (v_{i1}, \ldots, v_{im})$ is drawn from a distribution $D_i$ over $\mathbb{R}^m$. We will further assume that all bidders are independent and that $D_1 = \cdots = D_n$. As a result V is drawn from $D := \otimes_{i=1}^n D_i = D_1^{\otimes^n}$.

**Definition 1.** *An auction is defined by a randomized allocation rule $g = (g_1, \ldots, g_n)$ and a payment rule $p = (p_1, \ldots, p_n)$ where $g_i \colon \mathbb{R}^{n \times m} \to [0, 1]^m$ and $p_i \colon \mathbb{R}^{n \times m} \to \mathbb{R}_{\geqslant 0}$. Additionally for all items $j$ and valuation profiles $V$, the $g_i$ must satisfy $\sum_i [g_i(V)]_j \leqslant 1$.*

Given a bid matrix $B = (b_{ij})_{i \in N, j \in M}$, $[g_i(B)]_j$ is the probability that bidder $i$ receives object $j$ and $p_i(B)$ is the price bidder $i$ has to pay to the auction. The condition $\sum_i [g_i(V)]_j \leqslant 1$ allows the possibility for an item to be not allocated.

**Definition 2.** *The utility of bidder $i$ is defined by $u_i(\vec{v}_i, B) = \sum_{j=1}^m [g_i(B)]_j v_{ij} - p_i(B)$.*

Bidders seek to maximize their utility and may report bids that are different from their true valuations. In the following, we will denote by $B_{-i}$ the $(n-1) \times m$ bid matrix without bidder $i$, and by $(\vec{b}_i', B_{-i})$ the $n \times m$ bid matrix that inserts $\vec{b}_i'$ into row $i$ of $B_{-i}$ (for example: $B := (\vec{b}_i, B_{-i})$). We aim at auctions that incentivize bidders to bid their true valuations.

**Definition 3.** *An auction $(g, p)$ is dominant strategy incentive compatible (DSIC) if each bidder's utility is maximized by reporting truthfully no matter what the other bidders report. For every bidder $i$, valuation $\vec{v}_i \in D_i$, bid $\vec{b}_i' \in D_i$ and bids $B_{-i} \in D_{-i}$, $u_i(\vec{v}_i, (\vec{v}_i, B_{-i})) \geqslant u_i(\vec{v}_i, (\vec{b}_i', B_{-i}))$.*

**Definition 4.** *An auction is individually rational (IR) if for all $i \in N$, $\vec{v}_i \in D_i$ and $B_{-i} \in D_{-i}$,*

$$u_i(\vec{v}_i, (\vec{v}_i, B_{-i})) \geqslant 0. \tag{IR}$$

In a DSIC auction, the bidders have the incentive to truthfully report their valuations and therefore, the revenue on valuation profile $V$ is $\sum_{i=1}^n p_i(V)$. Optimal auction design aims at finding a DSIC and IR auction that maximizes the expected revenue $rev := \mathbb{E}_{V \sim D}[\sum_{i=1}^n p_i(V)]$. Since there is no known characterization of DSIC mechanisms in the multi-item setting, we resort to the relaxed notion of *ex-post regret*. It measures the extent to which an auction violates DSIC.

**Definition 5.** *The ex-post regret for a bidder $i$ is the maximum increase in his utility when considering all his possible bids and fixing the bids of others. For a valuation profile $V$, it is given by $r_i(V) = \max_{\vec{b}_i' \in \mathbb{R}^m} u_i(\vec{v}_i, (\vec{b}_i', V_{-i})) - u_i(\vec{v}_i, (\vec{v}_i, V_{-i}))$. In particular, DSIC is equivalent to*

$$r_i(V) = 0, \ \forall i \in N, \forall V \in D. \tag{IC}$$

*The bid $\vec{b}_i'$ that achieves $r_i(V)$ is called the optimal misreport of bidder $i$ for valuation profile $V$.*

Therefore, finding an optimal auction is equivalent to the following linear program:

$$\min_{(g,p) \in \mathcal{M}} - \mathbb{E}_{V \sim D}\left[\sum_{i=1}^n p_i(V)\right] \quad \text{s.t.} \quad \begin{array}{ll} r_i(V) = 0, & \forall\, i \in N, \ \forall\, V \in D, \\ u_i(\vec{v}_i, (\vec{v}_i, B_{-i})) \geqslant 0, & \forall i \in N, \ \vec{v}_i \in D_i, B_{-i} \in D_{-i}. \end{array} \tag{LP}$$

## 2.2 AUCTION DESIGN AS A LEARNING PROBLEM

As the space of auctions $\mathcal{M}$ may be large, we will set a parametric model. In what follows, we consider the class of auctions $(g^w, p^w)$ encoded by a neural network of parameter $w \in \mathbb{R}^d$. The corresponding utility and regret function will be denoted by $u_i^w$ and $r_i^w$.

Following Duetting et al. (2019), the formulation (LP) is relaxed: the IC constraint for all $V \in D$ is replaced by the expected constraint $\mathbb{E}_{V \sim D}[r_i^w(V)] = 0$ for all $i \in N$. The justification for this relaxation can be found in Duetting et al. (2019). By replacing expectations with empirical averages, the learning problem becomes:

$$\min_{w \in \mathbb{R}^d} -\frac{1}{L} \sum_{\ell=1}^{L} \sum_{i=1}^{n} p_i^w(V^{(\ell)}) \quad \text{s.t.} \quad \widehat{r}_i^w := \frac{1}{L} \sum_{\ell=1}^{L} r_i^w(V^{(\ell)}) = 0, \ \forall i \in N. \tag{$\widehat{\text{LP}}$}$$

The learning problem ($\widehat{\text{LP}}$) does not ensure (IR). However, this constraint is usually built into the parametrization (architecture) of the model: by design, the only auction mechanism considered satisfy (IR). Implementation details can be found in Duetting et al. (2019); Rahme et al. (2021) or in Sec 4.

## 3 AUCTION LEARNING AS A TWO-PLAYER GAME

We first present the optimization and the training procedures for ($\widehat{\text{LP}}$) proposed by Duetting et al. (2019). We then demonstrate with numerical evidence that this approach presents two limitations: hyperparameter sensitivity and lack of interpretability. Using the concept of $\varepsilon$-truthful to truthful reductions, we construct a new loss function that circumvents these two aspects. Lastly, we resort to amortized optimization and reframe the auction learning problem as a two-player game.

### 3.1 THE AUGMENTED LAGRANGIAN METHOD AND ITS SHORTCOMINGS

**Optimization and training.** We briefly review the training procedure proposed by Duetting et al. (2019) to learn optimal auctions. The authors apply the augmented Lagrangian method to solve the constrained problem ($\widehat{\text{LP}}$) and consider the loss:

$$\mathcal{L}(w; \lambda; \rho) = -\frac{1}{L} \sum_{\ell=1}^{L} \sum_{i \in N} p_i^w(V^{(\ell)}) + \sum_{i \in N} \lambda_i r_i^w(V^{(\ell)}) + \frac{\rho}{2} \left( \sum_{i \in N} r_i^w(V^{(\ell)}) \right)^2,$$

where $\lambda \in \mathbb{R}^n$ is a vector of Lagrange multipliers and $\rho > 0$ is a parameter controlling the weight of the quadratic penalty. More details about the training procedure can be found in Appendix A.

**Scheduling consistency problem.** The parameters $\lambda$ and $\rho$ are time-varying. Indeed, their value changes according to a pre-defined scheduling of the following form: 1) Initialize $\lambda$ and $\rho$ with respectively $\lambda^0$ and $\rho^0$, 2) Update $\rho$ every $T_\rho$ iterations : $\rho^{t+1} \leftarrow \rho^t + c$, where $c$ is a pre-defined constant, 3) Update $\lambda$ every $T_\lambda$ iterations according to $\lambda_i^t \leftarrow \lambda_i^t + \rho^t \widehat{r}_i^{w^t}$.

Therefore, this scheduling requires to set up five hyper parameters $(\lambda^0, \rho^0, c, T_\lambda, T_\rho)$. Some of the experiments found Duetting et al. (2019) were about learning an optimal mechanism for an $n$-bidder $m$-item auction ($n \times m$) where the valuations are iid $\mathcal{U}[0, 1]$. Different scheduling parameters were used for different values of $n$ and $m$. We report the values of the hyper parameters used for the $1 \times 2$, $3 \times 10$ and $5 \times 10$ settings in Table 1(a). A natural question is whether the choice of parameters heavily affects the performance. We proceed to a numerical investigation of this questions by trying different schedulings (columns) for different settings (rows) and report our the results in Table 1(b).

Table 1: (a): Scheduling parameters values set in Duetting et al. (2019) to reach optimal auctions in $n \times m$ settings with $n$ bidders, $m$ objects and i.i.d. valuations sampled from $\mathcal{U}[0, 1]$. (b): Revenue $rev := \mathbb{E}_{V \sim D}[\sum_{i=1}^{n} p_i(V)]$ and average regret per bidder $reg := 1/n \, \mathbb{E}_{V \in D} \left[ \sum_{i=1}^{n} r_i(V) \right]$ for $n \times m$ settings when using the different parameters values set reported in (a).

|  | $1 \times 2$ | $3 \times 10$ | $5 \times 10$ |
| --- | --- | --- | --- |
| $\lambda^0$ | 5 | 5 | 1 |
| $\rho^0$ | 1 | 1 | 0.25 |
| c | 50 | 1 | 0.25 |
| $T_\lambda$ | $10^2$ | $10^2$ | $10^2$ |
| $T_\rho$ | $10^4$ | $10^4$ | $10^5$ |

(a)

| | Scheduling | | | | | |
| --- | --- | --- | --- | --- | --- | --- |
| | $1 \times 2$ | | $3 \times 10$ | | $5 \times 10$ | |
| Setting | rev | rgt | rev | rgt | rev | rgt |
| $1 \times 2$ | 0.552 | 0.0001 | 0.573 | 0.0012 | 0.332 | 0.0179 |
| $3 \times 10$ | 4.825 | 0.0007 | 5.527 | 0.0017 | 5.880 | 0.0047 |
| $5 \times 10$ | 4.768 | 0.0006 | 5.424 | 0.0033 | 6.749 | 0.0047 |

(b)

The auction returned by the network dramatically varies with the choice of scheduling parameters. When applying the parameters of $1 \times 2$ to $5 \times 10$, we obtain a revenue that is lower by 30%! The performance of the learning algorithm strongly depends on the specific values of the hyperparameters. Finding an adequate scheduling requires an extensive and time consuming hyperparameter search.

**Lack of interpretability.** How should one compare two mechanisms with different expected revenue and regret? Is a mechanism $M_1$ with revenue $P_1 = 1.01$ and an average total regret $R_1 = 0.02$ better than a mechanism $M_2$ with $P_2 = 1.0$ and $R_2 = 0.01$ ? The approach in Duetting et al. (2019) cannot answer this question. To see that, notice that when $\lambda_1 = \cdots = \lambda_n = \lambda$ we can rewrite $\mathcal{L}(w; \lambda; \rho) = -P + \lambda R + \frac{\rho}{2} R^2$. Which mechanism is better depends on the values of $\lambda$ and $\rho$. For example if $\rho = 1$ and $\lambda = 0.1$ we find that $M_1$ is better, but if $\rho = 1$ and $\lambda = 10$ then $M_2$ is better. Since the values of $\lambda$ and $\rho$ change with time, the Lagrangian approach in Duetting et al. (2019) cannot provide metric to compare two mechanisms.

### 3.2 A TIME-INDEPENDENT AND INTERPRETABLE LOSS FUNCTION FOR AUCTION LEARNING

Our first contribution consists in introducing a new loss function for auction learning that addresses the two first limitations of Duetting et al. (2019) mentioned in Section 3.1. We first motivate this loss in the one bidder case and then extend it to auctions with many bidders.

#### 3.2.1 MECHANISMS WITH ONE BIDDER

**Proposition 1.** *[Balcan et al. (2005), attributed to Nisan] Let $\mathcal{M}$ be an additive auction with 1 bidder and $m$ items. Let $P$ and $R$ denote the expected revenue and regret, $P = \mathbb{E}_{V \in D}[p(V)]$ and $R = \mathbb{E}_{V \in D}[r(V)]$. There exists a mechanism $\mathcal{M}^*$ with expected revenue $P^* = (\sqrt{P} - \sqrt{R})^2$ and zero regret $R^* = 0$.*

A proof of this proposition can be found in Appendix C. Comparing two mechanisms is straightforward when both of them have zero-regret: the best one achieves the highest revenue. Prop. 1 allows a natural and simple extension of this criteria for non zero-regret mechanism with one bidder: we will say that $M_1$ is better than $M_2$ if and only if $M_1^*$ is better than $M_2^*$:

$$M_1 \geqslant M_2 \iff P^*(M_1) \geqslant P^*(M_2) \iff \sqrt{P_1} - \sqrt{R_1} \geqslant \sqrt{P_2} - \sqrt{R_2}$$

Using our metric, we find that a one bidder mechanism with revenue of 1.00 and regret of 0.01 is "better" than one with revenue 1.01 and regret 0.02.

#### 3.2.2 MECHANISMS WITH MULTIPLE BIDDERS

Let $M_1$ and $M_2$ be two mechanisms with $n$ bidders and $m$ objects. Let $P_i$ and $R_i$ denote their total expected revenue and regret, $P_i = \mathbb{E}_{V \in D} \left[ \sum_{j=1}^{n} p_j(V) \right]$ and $R_i = \mathbb{E}_{V \in D} \left[ \sum_{j=1}^{n} r_j(V) \right]$. We can extend our metric derived in Section 3.2.1 to the multiple bidder by the following:

$$M_1 \text{ is "better" than } M_2 \iff M_1 \geqslant M_2 \iff \sqrt{P_1} - \sqrt{R_1} \geqslant \sqrt{P_2} - \sqrt{R_2}$$

When $n = 1$ we recover the criteria from Section 3.2.1 that is backed by Prop. 1. When $n > 1$, it is considered a major open problem whether the extension of Prop. 1 still holds. Note that a multi-bidder variant of Prop. 1 *does* hold under a different solution concept termed "Bayesian Incentive Compatible" (Rubinstein & Weinberg, 2018; Cai et al., 2019), supporting the conjecture that Prop. 1

indeed extends.[2] Independently of whether or not Prop. 1 holds, this reasoning implies a candidate loss function for the multi-bidder setting which we can evaluate empirically.

This way of comparing mechanisms motivates the use of loss function: $\mathcal{L}(P, R) = -(\sqrt{P} - \sqrt{R})$ instead of the Lagrangian from Section 3, and indeed this loss function works well in practice. We empirically find the loss function $\mathcal{L}_m(P, R) = -(\sqrt{P} - \sqrt{R}) + R$ further accelerates training, as it further (slightly) biases towards mechanisms with low regret. Both of these loss function are time-independent and hyperparameter-free.

### 3.3 AMORTIZED MISREPORT OPTIMIZATION

To compute the regret $r_i^w(V)$ one has to solve the optimization problem: $\max_{\vec{v}_i' \in \mathbb{R}^m} u_i^w(\vec{v}_i, (\vec{v}_i', V_{-i})) - u_i^w(\vec{v}_i, (\vec{v}_i, V_{-i}))$. In Duetting et al. (2019), this optimization problem is solved with an inner optimization loop for each valuation profile. In other words, computing the regret of each valuation profile is solved separately and independently, from scratch. If two valuation profiles are very close to each other, one should expect that the resulting optimization problems to have close results. We leverage this to improve training efficiency.

We propose to amortize this inner loop optimization. Instead of solving all these optimization problems independently, we will instead learn one neural network $M^\varphi$ that tries to predict the solution of all of them. $M^\varphi$ takes as entry a valuation profile and maps it to the optimal misreport:

$$M^\varphi : \begin{cases} \mathbb{R}^{n \times m} & \to \mathbb{R}^{n \times m} \\ V = [\vec{v}_i]_{i \in N} & \to \left[ \text{argmax}_{\vec{v}' \in D} u_i(\vec{v}_i, (\vec{v}', V_{-i})) \right]_{i \in N} \end{cases}$$

The loss $\mathcal{L}_r$ that $M^\varphi$ is trying to minimize follows naturally from that definition and is then given by: $\mathcal{L}_r(\varphi, w) = -\mathbb{E}_{V \in D} \left[ \sum_{i=1}^n u_i^w(\vec{v}_i, ([M^\varphi(V)]_i, V_{-i})) \right]$.

### 3.4 AUCTION LEARNING AS A TWO-PLAYER GAME

In this section, we combine the ideas from Sections 3.2 and 3.3 to obtain a new formulation for the auction learning problem as a two-player game between an Auctioneer with parameter $w$ and a Misreporter with parameter $\varphi$. The optimal parameters for the auction learning problem $(w^*, \varphi^*)$ are a Nash Equilibrium for this game.

The Auctioneer is trying to design a truthful (IC) and rational (IR) auction that maximizes revenue. The Misreporter is trying to maximize the bidders' utility, for the current auction selected by Auctioneer, $w$. This is achieved by minimizing the loss function $\mathcal{L}_r(\varphi, w)$ wrt to $\varphi$ (as discussed in Sec 3.3). The Auctioneer in turn maximizes expected revenue, for the current misreports as chosen by Misreporter. This is achieved by minimizing $\mathcal{L}_m(w, \varphi) = -(\sqrt{P^w} + \sqrt{R^{w,\varphi}}) + R^{w,\varphi}$ with respect to $w$ (as discussed in Sec 3.2). Here, $R^{w,\varphi}$ is an estimate of the total regret that auctioneer computes for the current Misreporter $\varphi$, $R^{w,\varphi} = \frac{1}{L} \sum_{\ell=1}^L \sum_{i \in N} (u_i^w(\vec{v}_i, ([M^\varphi(V)]_i, V_{-i})) - u_i^w(\vec{v}_i, (\vec{v}_i, V_{-i})))$. This game formulation can be summarized as follows:

$$\begin{cases} \text{Misreporter:} & \begin{cases} \textbf{loss: } \mathcal{L}_r(\varphi, w) \\ \textbf{parameter: } \varphi \end{cases} \\ \text{Auctioneer:} & \begin{cases} \textbf{loss: } \mathcal{L}_m(w, \varphi) \\ \textbf{parameter: } w \end{cases} \end{cases} \quad (G)$$

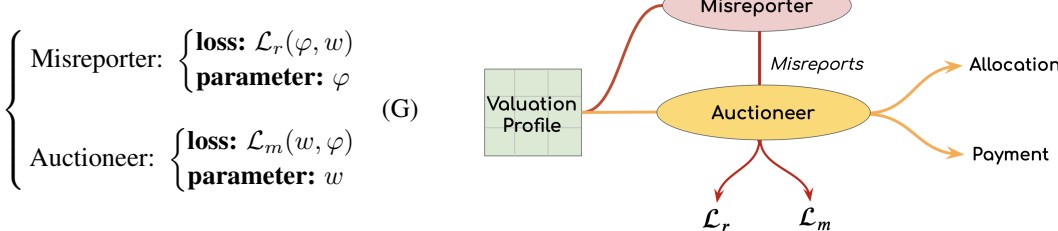

**Remark 1.** *The game formulation* (G) *reminds us of Generative Adversarial Networks (Goodfellow et al., 2014). Contrary to GANs, it is not a zero-sum game.*

---

[2] An auction is *Bayesian Incentive Compatible* if every bidder maximizes their expected utility by truthful reporting *in expectation over the other bidders' truthful bids*. Compare this to Dominant Strategy Incentive Compatible (our paper), where every bidder maximizes their expected utility by truthful reporting *for all realizations of the other bidders' bids*.

## 4 ARCHITECTURE AND TRAINING PROCEDURE

We describe ALGnet, a feed-forward architecture solving for the game formulation (G) and then provide a training procedure. ALGnet consists in two modules that are the auctioneer's module and the misreporter's module. These components take as input a bid matrix $B = (b_{i,j}) \in \mathbb{R}^{n \times m}$ and are trained jointly. Their outputs are used to compute the regret and revenue of the auction.

**Notation.** We use $\mathrm{MLP}(d_{\mathrm{in}}, n_l, h, d_{\mathrm{out}})$ to refer to a fully-connected neural network with input dimension $d_{\mathrm{in}}$, output dimension $d_{\mathrm{out}}$ and $n_l$ hidden layers of width $h$ and $\tanh$ activation function. sig denotes the sigmoid activation function. Given a matrix $B = [\vec{b}_1, \ldots, \vec{b}_n]^\top \in \mathbb{R}^{n \times m}$, we define for a fixed $i \in N$, the matrix $B_{(i)} := [\vec{b}_i, \vec{b}_1, \ldots, \vec{b}_{i-1}, \vec{b}_{i+1}, \ldots, \vec{b}_n]$.

### 4.1 THE AUCTIONEER'S MODULE

It is composed of an allocation network that encodes a randomized allocation $g^w \colon \mathbb{R}^{nm} \to [0,1]^{nm}$ and a payment network that encodes a payment rule $p^w \colon \mathbb{R}^{nm} \to \mathbb{R}^n$.

**Allocation network.** It computes the allocation probably of item $j$ to bidder $i$ $[g^w(B)]_{ij}$ as $[g^w(B)]_{ij} = [f_1(B)]_j \cdot [f_2(B)]_{ij}$ where $f_1 \colon \mathbb{R}^{n \times m} \to [0,1]^m$ and $f_2 \colon \mathbb{R}^{n \times m} \to [0,1]^{m \times n}$ are functions computed by two feed-forward neural networks.
  – $[f_1(B)]_j$ is the probability that object $j \in M$ is allocated and is given by $[f_1(B)]_j = \mathrm{sig}\,(\mathrm{MLP}(nm, n_a, h_a, n))$.
  – $[f_2(B)]_{ij}$ is the probability that item $j \in M$ is allocated to bidder $i \in N$ conditioned on object $j$ being allocated. A first MLP computes $l_j := \mathrm{MLP}(nm, n_a, h_a, m)(B_{(j)})$ for all $j \in M$. The network then concatenates all these vectors $l_j$ into a matrix $L \in \mathbb{R}^{n \times m}$. A softmax activation function is finally applied to $L$ to ensure feasibility i.e. for all $j \in M, \sum_{i \in N} L_{ij} = 1$.

**Payment network.** It computes the payment $[p^w(B)]_i$ for bidder $i$ as $[p^w(B)]_i = \tilde{p}_i \sum_{j=1}^m B_{ij}[g^w(B)]_{ij}$, where $\tilde{p} \colon \mathbb{R}^{n \times m} \to [0,1]^n$. $\tilde{p}_i$ is the fraction of bidder's $i$ utility that she has to pay to the mechanism. We compute $\tilde{p}_i = \mathrm{sig}\,(\mathrm{MLP}(nm, n_p, h_p, 1))\,(B_{(i)})$. Finally, notice that by construction $[p^w(B)]_i \leqslant \sum_{j=1}^m B_{ij} g^w(B)_{ij}$ which ensures that (IR) is respected.

### 4.2 THE MISREPORTER'S MODULE

The module consists in an $\mathrm{MLP}(nm, n_M, h_M, m)$ followed by a projection layer Proj that ensure that the output of the network is in the domain $D$ of the valuation. For example when the valuations are restricted to $[0,1]$, we can take $\mathrm{Proj} = \mathrm{sig}$, if they are non negative number,we can take $\mathrm{Proj} = \mathrm{SoftPlus}$. The optimal misreport for bidder $i$ is then given by $\mathrm{Proj} \circ \mathrm{MLP}(nm, n_M, h_M, m)(B_{(i)}) \in \mathbb{R}^m$. Stacking these vectors gives us the misreport matrix $M^\varphi(B)$.

### 4.3 TRAINING PROCEDURE AND OPTIMIZATION

We optimize the game (G) over the space of neural networks parameters $(w, \varphi)$. The algorithm is easy to implement (Alg. 1).

At each time $t$, we sample a batch of valuation profiles of size $B$. The algorithm performs $\tau$ updates for the Misreporter's network (line 9) and one update on the Auctioneer's network (line 10). Moreover, we often reinitialize the Misreporter's network every $T_{init}$ steps in the early phases of the training ($t \leqslant T_{limit}$). This step is not necessary but we found empirically that it speeds up training.

---

**Algorithm 1** ALGnet training

1: **Input**: number of agents, number of objects.
2: **Parameter**: $\gamma > 0$; $B, T, T_{init}, T_{limit}, \tau \in \mathbb{N}$.
3: **Initialize** misreport's and auctioneer's nets.
4: **for** $t = 1, \ldots, T$ **do**
5:     **if** $t \equiv 0 \bmod T_{init}$ and $t < T_{Limit}$ **then**:
6:         Reinitialize Misreport Network
7:     Sample valuation batch $S$ of size $B$.
8:     **for** $s = 1, \ldots, \tau$ **do**
9:         $\varphi^{s+1} \leftarrow \varphi^s - \gamma \nabla_\varphi \mathcal{L}_r(\varphi^s, w^t)(S)$.
10:     $w^{t+1} \leftarrow w^t - \gamma \nabla_w \mathcal{L}_m(w^t, \varphi)(S)$.

---

## 5 EXPERIMENTAL RESULTS

We show that ALGnet can recover near-optimal auctions for settings where the optimal solution is known and that it can find new auctions for settings where analytical solution are not known. Since RegretNet is already capable of discovering near optimal auctions, one cannot expect ALGnet to achieve significantly higher optimal revenue than RegretNet. The results obtained are competitive or better than the ones obtained in Duetting et al. (2019) while requiring much less hyperparameters (Section 3). We also evaluate ALGnet in online auctions and compare it to RegretNet.

For each experiment, we compute the total revenue $rev := \mathbb{E}_{V \sim D}[\sum_{i \in N} p_i^w(V)]$ and average regret $rgt := {}^1/n \, \mathbb{E}_{V \sim D}[\sum_{i \in N} r_i^w(V)]$ on a test set of $10,000$ valuation profiles. We run each experiment 5 times with different random seeds and report the average and standard deviation of these runs. In our comparisons we make sure that ALGnet and RegretNet have similar sizes for fairness (Appendix D).

## 5.1 AUCTIONS WITH KNOWN AND UNKNOWN OPTIMA

**Known settings.** We show that ALGnet is capable of recovering near optimal auction in different well-studied auctions that have an analytical solution. These are one bidder and two items auctions where the valuations of the two items $v_1$ and $v_2$ are independent. We consider the following settings. (A): $v_1$ and $v_2$ are i.i.d. from $\mathcal{U}[0,1]$, (B): $v_1 \sim \mathcal{U}[4,16]$ and $v_2 \sim \mathcal{U}[4,7]$, (C): $v_1$ has density $f_1(x) = 5/(1+x)^6$ and $v_2$ has density $f_2(y) = 6/(1+y)^7$.

(A) is the celebrated Manelli-Vincent auction (Manelli & Vincent, 2006); (B) is a non-i.i.d. auction and (C) is a non-i.i.d. heavy-tail auction and both of them are studied in Daskalakis et al. (2017). We compare our results to the theoretical optimal auction (Table 2). (Duetting et al. (2019) does not evaluate RegretNet on settings (B) & (C)). During the training process, $reg$ decreases to 0 while $rev$ and $P^*$ converge to the optimal revenue. For (A), we also plot $rev$, $rgt$ and $P^*$ as function of the number of epochs and we compare it to RegretNet (Fig. 1).

Contrary to ALGnet, we observe that RegretNet overestimates the revenue in the early stages of training at the expense of a higher regret. As a consequence, ALGnet learns the optimal auction faster than Regret-Net while being schedule-free and requiring less hyperparameters.

Table 2: Revenue & regret of ALGnet for settings (A)-(C).

|     | Optimal | | ALGnet (Ours) | |
| --- | --- | --- | --- | --- |
|     | $rev$ | $rgt$ | $rev$ | $rgt$ $(\times 10^{-3})$ |
| (A) | 0.550 | 0 | 0.555 ($\pm 0.0019$) | 0.55 ($\pm 0.14$) |
| (B) | 9.781 | 0 | 9.737 ($\pm 0.0443$) | 0.75 ($\pm 0.17$) |
| (C) | 0.1706 | 0 | 0.1712 ($\pm 0.0012$) | 0.14 ($\pm 0.07$) |

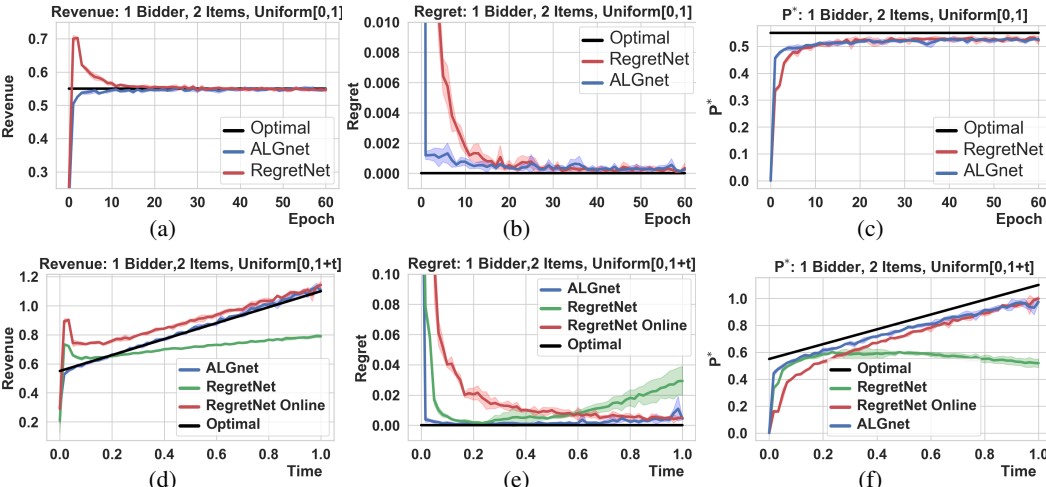

Figure 1: (a-b-c) compares the evolution of the revenue, regret and $P^*$ as a function of the number of epoch for RegretNet and ALGnet for setting (A). (d-e-f) plots the the revenue, regret and $P^*$ as a function of time for ALGnet and (offline & online) RegretNet for an online auction (Section 5.2).

**Unknown and large-scale auctions.** We now consider settings where the optimal auction is unknown. We look at $n$-bidder $m$-item additive settings where the valuations are sampled i.i.d from $\mathcal{U}[0,1]$ which we will denote by $n \times m$. In addition to "reasonable"-scale auctions ($1 \times 10$ and $2 \times 2$), we investigate large-scale auctions ($3 \times 10$ and $5 \times 10$) that are much more complex. Only deep learning methods are able to solve them efficiently. Table 3 shows that ALGnet is able to discover auctions that yield comparable or better results than RegretNet.

## 5.2 ONLINE AUCTIONS

ALGnet is an online algorithm with a time-independent loss function. We would expect it to perform well in settings where the underlying distribution of the valuations changes over time. We consider a one bidder and two items additive auction with valuations $v_1$ and $v_2$ sampled i.i.d from $\mathcal{U}[0, 1+t]$ where $t$ in increased from 0 to 1 at a steady rate. The optimal auction at time $t$ has revenue

Table 3: Comparison of RegretNet and ALGnet. The values reported for RegretNet are found in Duetting et al. (2019), the numerical values for $rgt$ and standard deviations are not available.

| Setting | RegretNet | | ALGnet (Ours) | |
|---|---|---|---|---|
| | $rev$ | $rgt$ | $rev$ | $rgt$ |
| $1 \times 2$ | 0.554 | $< 1.0 \cdot 10^{-3}$ | 0.555 $(\pm 0.0019)$ | $0.55 \cdot 10^{-3} (\pm 0.14 \cdot 10^{-3})$ |
| $1 \times 10$ | 3.461 | $< 3.0 \cdot 10^{-3}$ | 3.487 $(\pm 0.0135)$ | $1.65 \cdot 10^{-3} (\pm 0.57 \cdot 10^{-3})$ |
| $2 \times 2$ | 0.878 | $< 1.0 \cdot 10^{-3}$ | 0.879 $(\pm 0.0024)$ | $0.58 \cdot 10^{-3} (\pm 0.23 \cdot 10^{-3})$ |
| $3 \times 10$ | 5.541 | $< 2.0 \cdot 10^{-3}$ | 5.562 $(\pm 0.0308)$ | $1.93 \cdot 10^{-3} (\pm 0.33 \cdot 10^{-3})$ |
| $5 \times 10$ | 6.778 | $< 5.0 \cdot 10^{-3}$ | 6.781 $(\pm 0.0504)$ | $3.85 \cdot 10^{-3} (\pm 0.43 \cdot 10^{-3})$ |

$0.55 \times (1 + t)$. We use ALGnet and two versions of RegretNet, the original offline version (Appendix A) and our own online version (Appendix B) and plot $rev(t)$, $rgt(t)$ and $P^*(t)$ (Fig. 1). The offline version learns from a fixed dataset of valuations sampled at $t = 0$ (i.e. with $V \sim \mathcal{U}[0,1]^{nm}$) while the online versions (as ALGnet) learns from a stream of data at each time $t$. Overall, ALGnet performs better than the other methods. It learns an optimal auction faster at the initial (especially compared to RegretNet Online) and keep adapting to the distributional shift (contrary to vanilla RegretNet).

## 6 CONCLUSION

We identified two inefficiencies in previous approaches to deep auction design and propose solutions, building upon recent trends and results from machine learning (amortization) and theoretical auction design (stationary Lagrangian). This resulted in a novel formulation of auction learning as a two-player game between an Auctioneer and a Misreporter and a new architecture ALGnet. ALGnet requires significantly fewer hyperparameters than previous Lagrangian approaches. We demonstrated the effectiveness of ALGnet on a variety of examples by comparing it to the theoretical optimal auction when it is known, and to RegretNet when the optimal solution is not known.

**Acknowledgements.** Jad Rahme would like to thank Ryan P. Adams for helpful discussions and feedback on the manuscript. Samy Jelassi thanks Arthur Mensch for fruitful discussions on the subject and feedback on the manuscript. The work of Jad Rahme was funded by a Princeton SEAS Innovation Grant. The work of Samy Jelassi is supported by the NSF CAREER CIF 1845360. The work of S. Matthew Weinberg was supported by NSF CCF-1717899.

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

# A   TRAINING ALGORITHM FOR REGRET NET

We present the training algorithm for RegretNet, more details can be found in Duetting et al. (2019).

---

**Algorithm 2** Training Algorithm.

---

1: **Input**: Minibatches $\mathcal{S}_1, \ldots, \mathcal{S}_T$ of size $B$
2: **Parameters**: $\gamma > 0$, $\eta > 0$, $c > 0$, $R \in \mathbb{N}$, $T \in \mathbb{N}$, $T_\rho \in \mathbb{N}$, $T_\lambda \in \mathbb{N}$.
3: **Initialize Parameters**: $\rho^0 \in \mathbb{R}$, $w^0 \in \mathbb{R}^d$, $\lambda^0 \in \mathbb{R}^n$,
4: **Initialize Misreports:** $v_i'^{(\ell)} \in \mathcal{D}_i$, $\forall \ell \in [B]$, $i \in N$.
5:
6: **for** $t = 0, \ldots, T$ **do**
7:   Receive minibatch $\mathcal{S}_t = \{V^{(1)}, \ldots, V^{(B)}\}$.
8:   **for** $r = 0, \ldots, R$ **do**
9:           $\forall \ell \in [B]$, $i \in n$ :

$$v_i'^{(\ell)} \leftarrow v_i'^{(\ell)} + \gamma \nabla_{v_i'} u_i^{w^t}(v_i^{(\ell)}; (v_i'^{(\ell)}, V_{-i}^{(\ell)}))$$

10:
11:   Get Lagrangian gradient and update $w^t$:
12:     $w^{t+1} \leftarrow w^t - \eta \nabla_w \mathcal{L}(w^t; \lambda^t; \rho^t)$.
13:
14:   Update $\rho$ once in $T_\rho$ iterations:
15:   **if** $t$ is a multiple of $T_\rho$ **then**
16:     $\rho^{t+1} \leftarrow \rho^t + c$
17:   **else**
18:     $\rho^{t+1} \leftarrow \rho^t$
19:
20:   Update Lagrange multipliers once in $T_\lambda$ iterations:
21:   **if** $t$ is a multiple of $T_\lambda$ **then**
22:     $\lambda_i^{t+1} \leftarrow \lambda_i^t + \rho^t \, \widehat{r}_i(w^t), \forall i \in N$
23:   **else**
24:     $\lambda^{t+1} \leftarrow \lambda^t$

---

# B    TRAINING ALGORITHM FOR ONLINE REGRET NET

We present an online version of the training algorithm for RegretNet, more details can be found in Duetting et al. (2019). This version in mentionned in the orginal paper but the algorithm is not explicitly written there. The following code is our own adaptation of the original RegretNet algorithm for online settings.

---

**Algorithm 3** Training Algorithm.

---

1: **Input**: Valuation's Distribution $\mathcal{D}$
2: **Parameters**: $\gamma > 0$, $\eta > 0$, $c > 0$, $R \in \mathbb{N}$, $T \in \mathbb{N}$, $T_\rho \in \mathbb{N}$, $T_\lambda \in \mathbb{N}$, $B \in \mathbb{N}$
3: **Initialize Parameters**: $\rho^0 \in \mathbb{R}$, $w^0 \in \mathbb{R}^d$, $\lambda^0 \in \mathbb{R}^n$,
4: **for** $t = 0, \dots, T$ **do**
5:     Sample minibatch $\mathcal{S}_t = \{V^{(1)}, \dots, V^{(B)}\}$ from distribution $\mathcal{D}$.
6:     Initialize Misreports: $v_i'^{(\ell)} \in \mathcal{D}_i$, $\forall \ell \in [B]$, $i \in N$.
7:
8:     **for** $r = 0, \dots, R$ **do**
9:                             $\forall \ell \in [B]$, $i \in n$ :
$$v_i'^{(\ell)} \leftarrow v_i'^{(\ell)} + \gamma \nabla_{v_i'} u_i^{w^t}(v_i^{(\ell)}; (v_i'^{(\ell)}, V_{-i}^{(\ell)}))$$
10:
11:     Get Lagrangian gradient and update $w^t$:
12:         $w^{t+1} \leftarrow w^t - \eta \nabla_w \mathcal{L}(w^t; \lambda^t; \rho^t)$.
13:
14:     Update $\rho$ once in $T_\rho$ iterations:
15:     **if** $t$ is a multiple of $T_\rho$ **then**
16:         $\rho^{t+1} \leftarrow \rho^t + c$
17:     **else**
18:         $\rho^{t+1} \leftarrow \rho^t$
19:
20:     Update Lagrange multipliers once in $T_\lambda$ iterations:
21:     **if** $t$ is a multiple of $T_\lambda$ **then**
22:         $\lambda_i^{t+1} \leftarrow \lambda_i^t + \rho^t \widehat{r}_i(w^t), \forall i \in N$
23:     **else**
24:         $\lambda^{t+1} \leftarrow \lambda^t$

---

## C   Proof of Prop. 1

**Lemma 1.** *Let $M$ be a one bidder $m$ item mechanism with expected revenue $P$ and expected regret $R$, then $\forall \varepsilon > 0$, there exists a mechanism $M'$ with expected revenue $P' = (1-\varepsilon)P - \frac{1-\varepsilon}{\varepsilon}R$ and zero expected regret, $R' = 0$.*

*Proof.* For every valuation vector $v \in D$, let $g(v)$ and $p(v)$ denote the allocation vector and price that $M$ assigns to $v$.

We now consider the mechanism $M'$ that does the following:

- $g'(v) = g(v')$

- $p'(v) = (1-\varepsilon)\, p(v')$

Where $v'$ is given by : $v' = \operatorname{argmax}_{\tilde{v} \in D} \langle v,\, g(\tilde{v}) \rangle - (1-\varepsilon)\, p(\tilde{v})$. By construction, the mechanism $M'$ has zero regret, all we have to do now is bound its revenue. If we denote by $R(v)$ the regret of the profile $v$ in the mechanism $M$, $R(v) = \max_{\tilde{v} \in D} \langle v,\, g(\tilde{v}) - g(v) \rangle - (p(\tilde{v}) - p(v))$ we have.

$$\langle v,\, g(v') \rangle - p(v') = \langle v,\, g(v) \rangle - p(v) + \langle v,\, g(v') - g(v) \rangle - (p(v') - p(v))$$
$$\leqslant \langle v,\, g(v) \rangle - p(v) + R(v)$$

Which we will write as:

$$\langle v,\, g(v) \rangle - p(v) \geqslant \langle v,\, g(v') \rangle - p(v') - R(v)$$

Second, we have by construction:

$$\langle v,\, g(v') \rangle - (1-\varepsilon)p(v') \geqslant \langle v,\, g(v) \rangle - (1-\varepsilon)p(v)$$

By summing these two relations we find :

$$p(v') \geqslant p(v) - \frac{R(v)}{\varepsilon}$$

Finally we get that:

$$p'(v) \geqslant (1-\varepsilon)\, p(v) - \frac{1-\varepsilon}{\varepsilon}\, R(v)$$

Taking the expectation we get:

$$P' \geqslant (1-\varepsilon)\, P - \frac{1-\varepsilon}{\varepsilon}\, R$$

$\square$

**Proposition 1.** *Let $\mathcal{M}$ be an additive auction with $1$ bidders and $m$ items. Let $P$ and $R$ denote the total expected revenue and regret, $P = \mathbb{E}_{V \in D}\left[p(V)\right]$ and $R = \mathbb{E}_{V \in D}\left[r(V)\right]$. There exists a mechanism $\mathcal{M}^*$ with expected revenue $P^* = \left(\sqrt{P} - \sqrt{R}\right)^2$ and zero regret $R^* = 0$.*

*Proof.* From Lemma 1 we know that $\forall \varepsilon > 0$, we can find a zero regret mechanism with revenue $P' = (1-\varepsilon)\, P - \frac{1-\varepsilon}{\varepsilon}\, R$. By optimizing over $\varepsilon$ we find that the best mechanism is the one correspond to $\varepsilon = \sqrt{\frac{R}{P}}$. The resulting optimal revenue is given by:

$$P^* = (1 - \sqrt{\frac{R}{P}})P - \frac{\sqrt{\frac{R}{P}}}{\sqrt{\frac{R}{P}}}R = P - 2\sqrt{PR} + R = \left(\sqrt{P} - \sqrt{R}\right)^2$$

$\square$

# D    IMPLEMENTATION AND SETUP

We implemented ALGnet in PyTorch and all our experiments can be run on Google's Colab plateform (with GPU). In Alg. 1, we used batches of valuation profiles of size $B \in \{500\}$ and set $T \in \{160000, 240000\}$, $T_{limit} \in \{40000, 60000\}$, $T_{init} \in \{800, 1600\}$ and $\tau \in \{100\}$.

We used the AdamW optimizer (Loshchilov & Hutter, 2017) to train the Auctioneer's and the Misreporter's networks with learning rate $\gamma \in \{0.0005, 0.001\}$. Typical values for the architecture's parameters are $n_a = n_p = n_m \in [3, 7]$ and $h_p = h_n = h_m \in \{50, 100, 200\}$. These networks are similar in size to the ones used for RegretNet in Duetting et al. (2019).

For each experiment, we compute the total revenue $rev := \mathbb{E}_{V \sim D}[\sum_{i \in N} p_i^w(V)]$ and average regret $rgt := 1/n \, \mathbb{E}_{V \sim D}[\sum_{i \in N} r_i^w(V)]$ using a test set of $10,000$ valuation profiles. We run each experiment 5 times with different random seeds and report the average and standard deviation of these runs.

