# OpenReview forum: "Auction Learning as a Two-Player Game"
_ICLR.cc/2021/Conference — ICLR 2021 Poster_

### Official Review · AnonReviewer3 · 2020-10-21
**Review of Auction Learning as a Two-Player Game**

**Rating:** 6
**Confidence:** 3

**Review:**

Objective of the paper:
The objective of the paper is to show that auction design can be views as a two-player game;  improving on past work, they provide schemes that give better performance and better time for learning.

Strong Points:
1)  The paper seems to offer improvements on recent past work.
2)  The paper provides a clear background on auction theory relevant to the problem.
3)  The paper delivers on the abstract.
4)  Some code is provided.

Weak Points:
1)  It is not clear that the improvement over past work is large.
2)  The improvements/framework seem to rest on unproven assumptions, e.g., the use of the loss function in section 3.2.2, and the discussion of "closeness" in section 3.3.  This makes the paper somewhat unclear in terms of what it is offering -- heuristic approaches that improve (empirically) on the previous work?  If so, what are the limitations?  Are there situations the heuristic might be troublesome?  (Can you emphasize more clearly that your results are heuristic in nature, albeit based on theoretical formulations.)

Overall Rating:  I think this is an interesting paper.  I think the heuristics proposed offer benefits empirically and are well grounded, but the authors could do a better job clarifying the possible weak points of this heuristic approach compared to previous work.  The result is perhaps of interest to a specialized audience (people interested in auctions);  it's not clear that more general applications of the techniques are available.

Questions for Authors:
I was confused by Table 2 where you describe comparing to the "optimal" auction that has lower revenue than your auction results in 2 of 3 cases.  Is "optimal" here just signifying zero-regret?  Or is something else going on?

It is not clear to this reviewer that the auctions chosen are representative in any way -- I assume they've been used in previous works or are otherwise standard?

Other Feedback:  The paper is a bit confusing at times, but I think that is because the authors were forced to keep descriptions short in order to fit within page limits.  I think going back and offering a bit more description for a longer version would be useful.  Overall though the writing is fine.

---

> ### Author Response · Authors · 2020-11-14
> **Author response 1/2**
>
> Thank you for your feedback and for giving the opportunity to clarify some points in our paper.  This is the first part of a two part answer.
>
> > It is not clear that the improvement over past work is large.
>
> RegretNet, given enough capacity, training time, and optimal hyper parameters is capable of finding a (near) optimal auction. We shouldn't expect any method to significantly outperform the revenue/regret of a carefully tuned and optimized RegretNet. The performance of RegretNet is very hyper parameter sensitive. If the value of these hyperparameters are not carefully tuned, Regret can fail to find an optimal auction. This is discussed in Section 3.1 and the results are reported in Table 1.   Our contribution with ALGnet is to provide a novel formulation for the auction learning problem as well as a more robust learning algorithm that requires much less hyper parameter tuning.
>
>
> > The improvements/framework seem to rest on unproven assumptions, e.g., the use of the loss function in section 3.2.2, and the discussion of "closeness" in section 3.3. This makes the paper somewhat unclear in terms of what it is offering -- heuristic approaches that improve (empirically) on the previous work? If so, what are the limitations? Are there situations the heuristic might be troublesome? (Can you emphasize more clearly that your results are heuristic in nature, albeit based on theoretical formulations.)
>
> Indeed, we propose a heuristic approach that improves (empirically) on the previous work.  To be more specific,
> - The main justification for our approach is experimental, and we provide strong experimental justification for this metric (in all cases, for any number of bidders) in Section 5.
> - When n = 1, there is additionally strong theoretical justification, via Proposition 1.
> - When n > 1, there is no theoretical justification, and we are explicit in 3.2.2 that we are not claiming any, and that “independently of whether or not Proposition 1 holds, this reasoning implies a candidate loss function for the multi-bidder setting which we can evaluate empirically.”
> - The comments we made about BIC are intended just to give context/related work. No results in the paper concern BIC, and we’re sorry if this caused a confusion.
>
> On the settings we considered, which include the ones reported in the paper but also settings that we did not report, our heuristic worked and we were able to learn an (near) optimal auction.
>
> >Overall Rating: I think this is an interesting paper. I think the heuristics proposed offer benefits empirically and are well grounded, but the authors could do a better job clarifying the possible weak points of this heuristic approach compared to previous work. The result is perhaps of interest to a specialized audience (people interested in auctions); it's not clear that more general applications of the techniques are available.
>
> While our results might be most relevant to people interested in Deep Learning and Auction design,  we believe that  the game formulation we derive could also be of interest to the broader ML community.
>
> ALGnet was able to learn optimal auction in all of the settings we tested it on with very little tuning. Game formulations however,  are known to be hard to train and could suffer from pathologies such as mode collapse (which is a common problem for GANs). While we did not observe such shortcomings in our experiments, we cannot exclude that ALGnet could suffer from such pathologies in possibly more complicated settings.

---

> > ### Author Response · Authors · 2020-11-14
> > **Author response 2/2**
> >
> > > Questions for Authors: I was confused by Table 2 where you describe comparing to the "optimal" auction that has lower revenue than your auction results in 2 of 3 cases. Is "optimal" here just signifying zero-regret? Or is something else going on?
> >
> > The auctions in Table 2, are auctions for which the optimal solution is known (as in derived mathematically). The optimal revenue is the highest revenue an auction with zero-regret can achieve. ALGnet gets a higher number because it is not exactly incentive compatible, the regret is not exactly zero.
> >
> >  Interestingly, if you compute the theoretical optimal revenue that we can get from the auction through the reduction: $P^{*} = (\sqrt{P} - \sqrt{R})^2$,  you can actually notice that it is lower than the theoretical optimal revenue. Please find below a table with these numbers:
> >
> > |             	|&nbsp;&nbsp;&nbsp;&nbsp;&nbsp;&nbsp; *ALGnet Revenue* &nbsp;&nbsp;&nbsp;&nbsp;&nbsp;&nbsp; 	|   &nbsp;&nbsp;&nbsp;&nbsp;&nbsp;&nbsp; *ALGnet Regret* &nbsp;&nbsp;&nbsp;&nbsp;&nbsp;&nbsp;|&nbsp;&nbsp;&nbsp;&nbsp;&nbsp;&nbsp;&nbsp;&nbsp;&nbsp;&nbsp;&nbsp;&nbsp;&nbsp;   $P^{*}$  	&nbsp;&nbsp;&nbsp;&nbsp;&nbsp;&nbsp;&nbsp;&nbsp;&nbsp;&nbsp;&nbsp;| &nbsp;&nbsp;&nbsp;&nbsp;&nbsp;&nbsp;  *Optimal Revenue* &nbsp;&nbsp;&nbsp;&nbsp;&nbsp;&nbsp;	|
> > |:----------------------------------------------------------------	|:----------------------------------------------------------------:|:----------------------------------------------------------------:|:----------------------------------------------------------------:|:-----------------------------------------------------------------:|
> > | *Setting (A)* 	|      $0.555$              	|     $0.55  \times 10^{-3}$      	|    $0.521$  	|    $0.55$           	|
> > | *Setting (B)* 	|      $9.737$              	|     $0.75 \times  10^{-3}$      	|    $9.567$  	|    $9.781$          	|
> > | *Setting (C)* 	|      $0.1712$             	|     $0.14  \times  10^{-3}$     	|   $0.1615$ 	|    $0.1706$         	|
> >
> > > It is not clear to this reviewer that the auctions chosen are representative in any way -- I assume they've been used in previous works or are otherwise standard?
> >
> > Indeed, these are standard auctions that have been studied and used in previous work.
> >
> > >Other Feedback: The paper is a bit confusing at times, but I think that is because the authors were forced to keep descriptions short in order to fit within page limits. I think going back and offering a bit more description for a longer version would be useful. Overall though the writing is fine.
> >
> > Having an extra page would give us enough space to add some clarifications and would be very helpful. Please let us know if there are other parts of the paper that you found confusing and we would be happy to clarify them.

---

### Official Review · AnonReviewer2 · 2020-10-22
**Nice paper! I have a question about relation to other work.**

**Rating:** 6
**Confidence:** 3

**Review:**

This paper revisits a recent deep learning framework by Duetting et al (ICML’19) for learning Bayesian optimal auctions, which is a notoriously hard problem in theory. They propose two modifications to the Duetting et al architecture:
•	The auctions computed by these algorithms are not exactly IC. To cope with that the learning objective penalizes by the regret of each agent from truthfully reporting their type. This leads to a question of choosing the relative weights of the revenue and the regret. Previous work had a complicated and brittle way of doing it, and the new paper proposes a simple formula that is inspired by auction theory and seems to work well in practice. The authors also propose to use this formula as a benchmark for comparing auctions with different revenue and regret guarantees.
•	The training algorithm has to estimate the regret at each step, which requires an expensive optimization for the optimal deviation. The new paper proposes to amortize this optimization: after each gradient step of the auction, the optimal deviation probably did not change much so we can start it from the previous optimal deviation.


This paper makes nice contributions in an exciting new research direction towards insights on a classical problem.


I’m still concerned about a couple of issues:
•	I don’t understand the relationship between this paper and Rahme et al. (“A Permutation-Equivariant Neural Network Architecture For Auction Design”): It also mentioned in passing in the introduction. I’m not familiar with the details of Rahme et al., but based on the abstract it seems like both propose improvements to the same Duetting et al. paper in the same setting. Many parts, e.g. structure of the introduction, are almost identical. Moreover, Rahme et al. claim to have a better algorithm than Duetting et al., so I’m not sure why the current paper doesn’t compare performance to that algorithm. (If these were written by the same authors, I’m not sure why they wrote 2 different papers?)
•	The authors propose that the same theory-inspired formula they used for balancing regret and revenue be used to compare different mechanisms with different levels of revenue/regret. For multi-agent settings this approach sounds like a plausible heuristic, but they don’t bring enough support to convince me to use it as a metric. For single-agent settings, I would advocate to just apply the usual reduction to revelation principle, i.e. ask how much revenue the mechanism would get with a rational agent who exactly optimizes their bid.

---

> ### Author Response · Authors · 2020-11-14
> **Author response**
>
> We would like to thank the reviewer for their feedback and suggestion.
>
> > I’m still concerned about a couple of issues: • I don’t understand the relationship between this paper and Rahme et al. (“A Permutation-Equivariant Neural Network Architecture For Auction Design”): It also mentioned in passing in the introduction. I’m not familiar with the details of Rahme et al., but based on the abstract it seems like both propose improvements to the same Duetting et al. paper in the same setting. Many parts, e.g. structure of the introduction, are almost identical. Moreover, Rahme et al. claim to have a better algorithm than Duetting et al., so I’m not sure why the current paper doesn’t compare performance to that algorithm. (If these were written by the same authors, I’m not sure why they wrote 2 different papers?)
>
> The present work and Rahme et al. (2020) both improve on RegretNet but not in the same way.
>
> Rahme et al. (2020) specializes in auctions that are bidder-and-item symmetric. They prove that such auctions have an optimal solution that is equivariant and consequently they build a neural network architecture (EquivariantNet) that can only represent those equivariant auctions.  They show that EquivariantNet is capable of learning optimal auctions just like RegretNet while enjoying better generalization properties (out of sample and out of setting).  These better generalization properties are a consequence of restricting the search space to equivariant auctions.
>
> Since RegretNet is capable of learning optimal auctions, EquivariantNet does not (and cannot) significantly improve over RegretNet in terms of higher revenue and lower regret. In fact, the revenue and regret reported in Rahme et al. (2020) show that with enough samples, EquivariantNet matches the performance of RegretNet. For this reason, we believe that a comparison with RegretNet is sufficient here and do not include a direct comparison of ALGnet with EquivariantNet.
> Our current submission does not propose a new neural network architecture. We are using fully connected neural networks to represent the allocation and payment mechanisms. As a result we are capable of learning very general auctions (similarly to RegretNet) and are not limited to symmetric auctions which is the case for EquivariantNet.  ALGnet can be applied to a wider range of settings.
>
> EquivariantNet, is trained using a similar procedure to RegretNet (the augmented Lagrangian method) and as a result still suffers from the same hyperparameter sensitivity, contrary to ALGnet.  ALGnet improves over RegretNet and EquivariantNet by proposing a more robust training algorithm that requires less hyperparameters.
> We would also like to mention that one of the contributions is also proposing a novel formulation of auction learning as a two player game between an Auctioneer and a Misreporter which is not present in prior work (including Duetting et al. 2019 and Rahme et al. 2020).
>
> Duetting et al. (2019), Rahme et al. (2020) and the current submission are working within the same framework: learning incentive compatible, individually rational additive auctions that maximizes revenue.  The definitions and notations needed to set up the problem are very similar.  We remind them in Section 2 to make the paper self contained.
>
>
>
>
> > The authors propose that the same theory-inspired formula they used for balancing regret and revenue be used to compare different mechanisms with different levels of revenue/regret. For multi-agent settings this approach sounds like a plausible heuristic, but they don’t bring enough support to convince me to use it as a metric. For single-agent settings, I would advocate to just apply the usual reduction to revelation principle, i.e. ask how much revenue the mechanism would get with a rational agent who exactly optimizes their bid.
>
>
> For the multi-bidder setting, we wish to briefly remind the reader that all our support is empirical: when we use this loss function, we find optimal auctions. For the single-bidder setting, the metric proposed by the reviewer is also good.
>
> Please let us know if you have other concerns, we would be happy to address them.

---

> > ### Comment · AnonReviewer2 · 2020-11-16
> > **Comparisons to EquivariantNet on symmetric examples?**
> >
> > Thanks for the interesting response!
> >
> > Your explanation of how EquivariantNet works only for symmetric priors is helpful!
> > However most/all of your experimental comparisons to AlgNet are on symmetric priors.
> > How does RegretNet compare to AlgNet on those instances?
> > Also, can you combine both insights?

---

> > > ### Author Response · Authors · 2020-11-17
> > > **Comparison with EquivariantNet**
> > >
> > > > Your explanation of how EquivariantNet works only for symmetric priors is helpful!
> > >
> > > Thank you!
> > >
> > > > However most/all of your experimental comparisons to AlgNet are on symmetric priors. How does RegretNet compare to AlgNet on those instances?
> > >
> > > In the following, we assume that you meant "How does *EquivariantNet* compare to AlgNet on those instances?" (and not RegreNet).
> > >
> > > We start with a quantitative comparison:
> > >
> > > |             	| &nbsp;&nbsp;&nbsp;&nbsp;&nbsp;&nbsp;*ALGnet Revenue* &nbsp;&nbsp;&nbsp;&nbsp;&nbsp;&nbsp;	| &nbsp;&nbsp;&nbsp;&nbsp;&nbsp;&nbsp;*ALGnet Regret* &nbsp;&nbsp;&nbsp;&nbsp;&nbsp;&nbsp;	|&nbsp;&nbsp;&nbsp;&nbsp;&nbsp;&nbsp; *EquivariantNet Revenue* &nbsp;&nbsp;&nbsp;&nbsp;&nbsp;&nbsp;	| &nbsp;&nbsp;&nbsp;&nbsp;&nbsp;&nbsp; *EquivariantNet Regret* &nbsp;&nbsp;&nbsp;&nbsp;&nbsp;&nbsp;	|&nbsp;&nbsp;&nbsp;&nbsp;&nbsp;&nbsp;  *Optimal Revenue* &nbsp;&nbsp;&nbsp;&nbsp;&nbsp;&nbsp;	|
> > > |:-------------	|:----------------:	|:---------------:	|:------------------------:	|:-----------------------:	|:----------------:	|
> > > | *Setting (A)* 	| $0.555$          	| $0.55 \times 10^{-3}$  	| $0.551$                  	| $0.13\times 10^{-3}$          	| $0.55$           	|
> > > | *Setting (B)* 	| $9.737$          	| $0.75 \times 10^{-3}$  	| $9.296$                  	| $18.5 \times 10^{-3}$          	| $9.781$          	|
> > > | *Setting (C)* 	| $0.1712$         	| $0.14 \times 10^{-3}$  	| $0.173$                  	| $0.03 \times 10^{-3}$          	| $0.1706$         	|
> > > | $2 \times 2$       	| $0.879$          	| $0.58 \times 10^{-3}$  	| $0.873$                  	| $1.0 \times 10^{-3}$           	| Unkown         	|
> > >
> > >
> > >
> > >
> > > - In this table, the values for settings (A), Setting (C) and  Setting ($2 \times 2$) are taken from Rahme et al. (2020).
> > > - Remark: Setting (A) is the same as  $1 \times 2$.
> > > - In addition to these experiments, following up on your question,  we ran an experiment on Setting (B) for EquivariantNet.  Setting (B) is interesting because it is a non symmetric auction (just like Setting C) for which the optimal solution is non symmetric (unlike Setting C).
> > >
> > > EquivariantNet and  ALGnet have comparable performances on symmetric auctions ( Setting A and $2 \times 2$) or non symmetric auction for which the optimal auction is known to be symmetric ( Setting C).  We can see that EquivariantNet can have a smaller regret than ALGnet (2/3 cases). This is to be expected because EquivariantNet limits itself to symmetric auctions (the search space is much smaller).
> > >
> > > On Setting (B) however, we can see that EquivariantNet is not as good a ALGnet.  EquivariantNet generates less revenue than ALGnet and has a regret that is  $25\times$  higher.  This is also to be expect because the optimal auction for that setting is not in the search space of EquivariantNet.
> > >
> > > From these experiments we see that ALGnet is competitive with EquivariantNet on symmetric auctions and better suited to learn non symmetric auctions.
> > >
> > > *Additional Comparaison:*
> > >  - EquivariantNet should be more sample efficient  than ALGnet on symmetric auctions. This is because EquivariantNet has a strong inductive prior built into its architecture (item-and-bidder equivariant auctions only).
> > > - ALGnet can learn larger set of auctions than EquivariantNet which includes non-symmetric auctions.
> > > - EquivariantNet is trained using the same training procedure as RegretNet. As a result, it suffers from the same disadvantages -  EquivariantNet it is very hyperparameter sensitive. ALGnet on the other hand, requires less hyperparameters and is more robust.
> > >
> > >
> > > > Also, can you combine both insights?
> > >
> > > Absolutely!  EquivariantNet improves over RegretNet on symmetric auctions by using a better suited architecture to learn these auctions.  ALGnet improves over RegretNet by making the training procedure more robust through a novel formulation. These two improvements are orthogonal to each other and it is perfectly possible to combine them.  We would just need to replace our current architecture (which is fully connected) by an equivariant one (similar to EquivariantNet). This would be very well suited for symmetric auctions (but also limited to these auctions).   More generally, we can combine any architecture-based improvement with ALGnet.
> > >
> > > Please let us know if we have answered your question and if you have additional ones.

---

### Official Review · AnonReviewer4 · 2020-10-26
**Optimal auction design using DNN as a two player game through hyper-parameter free and robust loss function is studied. The paper is not well placed in the literature, making the contributions hard to judge.**

**Rating:** 6
**Confidence:** 2

**Review:**

The authors study the problem of optimal auction design using deep neural networks. Based on the structural insights in optimal mechanism in an additive auction with 1 bidder and m items, they propose a new loss function. This loss function is hyper-parameter free unlike Duetting et al. (2019) which is one of the early works in this field, making it more robust and interpretable.
The authors then propose a training method that resembles a two-player game to optimize the new loss function. The synthetic experiments (Table 3) show the designed algorithm, which does not require hyperparameter tuning, is comparable to the work of Duetting et al. (2019).

Pros:
* The connection to the structural results in Balcan et al. (2005) to design the loss function.
* The illustration of the sensitivity of the loss function in Duetting et al. (2019) solidifies the message.
* The training method that resembles a two-player game brings a new perspective to the problem (*)

Cons/Questions:
* The paper lacks adequate comparisons, both conceptual and experimental, to other existing works.
* In the experiments section, for Table 3 how the hyperparameters for RegretNet (Duetting et al. (2019)) is tuned is not clear. Such tuning is crucial for RegretNet as mentioned by authors.
* In Table 2. comparison with RegretNet will be useful.
* In Fig 1. why the online RegretNet and ALGnet converge to the same solution, despite the different loss functions?
* What are the depth and the width of the neural networks used in the experiments?
* Is RegretNet the state-of-the-art for the different settings mentioned? (*)

Minor:
* The auctions reported have a maximum size of 5x10, which, in my opinion, should not be called a large scale.


(*) As an outsider to this field, whether this method is truly novel is hard for me to judge.

---

> ### Author Response · Authors · 2020-11-14
> **Author response**
>
> We would like to thank the reviewer for their feedback.
>
> >The training method that resembles a two-player game brings a new perspective to the problem (x)
>
> To the best of our knowledge, this is a novel formulation of the auction learning problem.
>
>
> > The paper lacks adequate comparisons, both conceptual and experimental, to other existing works.
>
> To the best of our knowledge, the main architecture that learns optimal auctions is RegretNet (Duetting et al. (2019)) for a general auction. This matches with the framework of our paper. For this reason, we make a thorough comparison (both conceptually and experimentally) with this architecture. Other existing works are more specialized and design architectures for specific economics problems (facility location problems (Golowich et al. (2018)) or make assumptions on the structure of the auction (Feng et al. (2018), Tacchetti et al (2019),  Rahme et al. (2020) ).
>
> Do you have a particular work in mind, you would like us to compare ALGnet to?
>
>
> > In the experiments section, for Table 3 how the hyperparameters for RegretNet (Duetting et al. (2019)) is tuned is not clear. Such tuning is crucial for RegretNet as mentioned by authors.
>
> You are right, RegretNet is very hyper-parameter sensitive. Tuning these hyperparameters is crucial to learn optimal auctions and ensure a fair comparison.  The numbers (revenue and regret) that are reported in Table 3 for RegretNet are taken from Duetting et al. (2019) as indicated in the caption of the table.  These values are optimal for RegretNet - they were found by authors of Duetting et al. (2019) through a hyperparameter search. Additionally, these values match the optimal values that we found for RegretNet through our own hyperparameter search using the code base that they released (https://github.com/saisrivatsan/deep-opt-auctions).  Consequently we think that the comparaison is fair.
>
>
> > In Table 2. comparison with RegretNet will be useful.
>
> Table 2 gathers auction settings where the optimal auction is analytically known. Therefore, we judged that it was not useful to add the results obtained by RegretNet. If the reviewer thinks that this comparison is important, we can add the result obtained with RegretNet on these auctions. Note that the first experiment (A) in Table 2 is the same as the first experiment from Table 3 so it is possible to find results obtained with RegretNet there for that setting.
>
>
> > In Fig 1. why the online RegretNet and ALGnet converge to the same solution, despite the different loss functions?
>
> Despite having different loss functions and training algorithms, both RegretNet and ALGnet are trying to learn an incentive compatible, individually rational, auction that maximizes revenue. Both architectures are capable of converging to the optimal auction and it is rather reassuring that the plots showcase that.
>
> > What are the depth and the width of the neural networks used in the experiments?
>
> Typical values for the depth and width of the Auctioneer and Misreporter Networks are reported in Appendix D.  Typical networks consist of 3-7 layers of width 50-200. These networks are similar in size to the ones used for RegretNet as found in the code they released (https://github.com/saisrivatsan/deep-opt-auctions)
>
>
> > Is RegretNet the state-of-the-art for the different settings mentioned? (x)
>
>
> RegretNet is state-of-the art in settings where the optimal auction is known as it is able to find this latter. In some settings where the optimal auction is not known, it is able to find auctions that are similar to the ones returned by other computational methods as LP solvers (Conitzer and Sandholm 2002, 2004). We insist on the fact that our contribution is not to obtain *significantly better* results than RegretNet in terms of revenue and regret. This is not possible,  RegretNet is capable of finding near optimal auctions, so is ALGnet. Instead, our contribution consists in finding a novel formulation of the auction learning problem and a new training algorithm that is more robust and requires less hyperparameters.
>
> >The auctions reported have a maximum size of 5x10, which, in my opinion, should not be called a large scale.
>
> It is important to note that settings where the optimal auction is analytically known consist in at most 2 bidders (with the exception of auctions with only one item).  Moreover, due to the exponential (in the number of bidders) number of constraints in the LP that is presented in Section 2, LP solvers are able to return an approximately optimal solution in more than one week for settings with 3 bidders (cf. Duetting et al. 2019 section 5.7). Therefore, auctions with 5 bidders and 10 items can already be considered as a “large scale” setting.
>
>
> Please let us know if you have any additional concerns.

---

> > ### Comment · AnonReviewer4 · 2020-11-20
> > **I have read the response and will increase my score**
> >
> > With the understanding that Duetting et al. (2019) presents the state-of-the-art algorithm in optimal auction design, I feel the current work presents a new perspective. Most of my other concerns are addressed adequately. Therefore, I have increased my score.
> >
> > The LP solvers can be presented with relaxed LPs in some settings to obtain near-optimal results, which may or may not work here. However, my issue was different. Why not run the current technique for instances larger than 10x5? Is that computationally prohibitive for the current algorithm (not LP solvers)?

---

> > > ### Author Response · Authors · 2020-11-21
> > > **Thank you for your feedback**
> > >
> > > Thank you for getting back to us and reconsidering your evaluation!
> > >
> > > It is possible to consider larger settings of the same order of magnitude, but substantially larger settings (100 bidders with 100 objects for example) are computationally prohibitive in terms of compute and GPU memory for current methods (ours and others).
> > >
> > > Please let us know if  you have any additional questions!

---

### Official Review · AnonReviewer1 · 2020-10-29
**A good empirical paper, which simplifies the training procedure proposed by the original paper on this topic [Duetting et al '19]. However, the authors need to clarify some overstatements in the paper. I think this paper can be accepted if there is room.**

**Rating:** 6
**Confidence:** 4

**Review:**

This paper studies the auction design project through deep learning, following the direction pioneered by [Duetting et al '19]. The main contribution of this paper lies in two aspects: 1) they propose a time-independent lagrangian loss function which is motivated from eps-BIC to BIC transformation in [Rubinstein & Weinberg '18], and 2) the authors propose a GAN structure to compute the regret during training. The idea is simple and clean: design another NN to approximately compute the optimal misreport for each bidder.

The good:

1. It lies in the intersection between Game Theory/Mechanism Design and Machine Learning, it should be of interest to a lot of people on both sides. It is a good fit for ICLR community.
2. It focuses on an important question in mechanism design literature: multi-dimensional revenue-maximizing auction design.
3. It is interesting and important that it can avoid hyper-parameter tuning for lagrangian loss function during training.

The drawbacks:

1. It follows the spirit of [Duetting et al.] and also inherits the drawback there: the trained mechanism is not strictly IC and uninterpretable.

2. The authors claim that they propose a metric to measure the performance of the nontruthful auctions, however, based on my understanding, it only holds for the single bidder case. For multiple bidders setting, as the authors state, they need to change to the BIC version of the regret, which is quite different with the expected ex post regret defined in this paper and the prior work [Duetting et al '19].  In this paper, all the simulations are based on the expected ex post regret. Therefore the new metric statement doesn't seem solid and the authors need to clarify.

In summary, this work is a good empirical paper, which simplifies the training procedure proposed by the original paper in [Duetting et al '19]. However, the authors need to clarify some statements in the paper. I think this paper can be accepted if there is room.

---

> ### Author Response · Authors · 2020-11-14
> **Author Response**
>
> Thank you for sharing your feedback and giving us an opportunity to clarify some of our statements.
>
> > It follows the spirit of [Duetting et al.] and also inherits the drawback there: the trained mechanism is not strictly IC and uninterpretable.
>
> That is true, the mechanisms we learn are not interpretable. This criticism also holds for most neural network models.  However, in low dimension, 1-bidder, 2-items for example, one can visualize the learned allocation rules and get a better understanding of what the mechanism is doing. This is done for example in Duetting et al. (2019) (Figure 4) for the 1-bidder, 2-item auction where the items valuations are iid Uniform[0,1].
>
> The reviewer is correct that the mechanisms are not strictly 0-IC when n > 1 (we briefly remind the reviewer that the reduction of Proposition 1 makes any mechanism 0-IC for n = 1). However, the regret can be made arbitrarily small, and this is standard in computational results in Auction Theory (see, e.g., Cai et al. 2012b, Cai et al. 2013a, Cai et al. 2013b, and the prior work we build from Duetting et al. 2019).
>
> >The authors claim that they propose a metric to measure the performance of the nontruthful auctions, however, based on my understanding, it only holds for the single bidder case. For multiple bidders setting, as the authors state, they need to change to the BIC version of the regret, which is quite different with the expected ex post regret defined in this paper and the prior work [Duetting et al '19]. In this paper, all the simulations are based on the expected ex post regret. Therefore the new metric statement doesn't seem solid and the authors need to clarify.
>
> Sorry that our wording was unclear in this section due to space limitations. Just to clarify:
> - We propose a metric in all cases, for any number of bidders. That metric is defined in 3.2.2, as $\mathcal{L}(P, R) = - (\sqrt{P} - \sqrt{R})$.
> - The main justification for this metric is experimental, and we provide strong experimental justification for this metric (in all cases, for any number of bidders) in Section 5.
> - When n = 1, there is additionally strong theoretical justification, via Proposition 1.
> - When n > 1, there is no theoretical justification, and we are explicit in 3.2.2 that we are not claiming any, and that “independently of whether or not Proposition 1 holds, this reasoning implies a candidate loss function for the multi-bidder setting which we can evaluate empirically.”
> - The comments we made about BIC are intended just to give context/related work. No results in the paper concern BIC, and we’re sorry for this confusion.
>
> > In summary, this work is a good empirical paper, which simplifies the training procedure proposed by the original paper in [Duetting et al '19]. However, the authors need to clarify some statements in the paper. I think this paper can be accepted if there is room.
>
> Is there any other statement /over-statement that you would like us to clarify?

---

### Official Review · AnonReviewer5 · 2020-11-04
**ICLR 2021 Conference Paper2740 AnonReviewer5**

**Rating:** 7
**Confidence:** 3

**Review:**

This paper studies how to design auctions using deep learning, as initiated by the work of Duetting et al. (2019). The authors argue that their new framework addresses several undesirability of the previous works, including extensive hyper-parameter tuning and unclear comparison between two auctions with different revenue and regret performance. Moreover, the new framework can be easily generalized to online environment.

The main novelty of this paper is to use a new optimization objective without relying on the Lagrangian method to formalize the objective. The new optimization objective is backed by a theorem stating that for a single-bidder environment, an auction with regret can be converted to an auction without regret with a revenue loss bounded by the square root of the regret, which also provides a convenient way to compare two auctions. Since the new objective does not consist of Lagrangian multipliers, no hyper-parameter tuning is required.

The empirical results further demonstrate the advantage of their new architectures.

Comments:

This paper is generally well-written and clear. The problem it studies is interesting and important, related to both game theory and deep learning, which is relevant and interesting to the deep learning community. It is nice to have a new architecture without extensive hyper-parameter tuning

1. The author mentions that the theorem (that for a single-bidder environment, an auction with regret can be converted to an auction without regret with a revenue loss bounded by the square root of the regret) can be extended to the multi-bidder environment if using the notion of BIC. However, according to the definition, the regret is still computed in the ex-post level, which seems a bit strange. Could you elaborate more on this?

2. Is it true that the misreport's utility is always 0 in any Nash equilibrium of the two-player game? Or this is only true for the optimal solution?

---

> ### Author Response · Authors · 2020-11-14
> **Author response**
>
> Thank you for reviewing our paper and sharing you feedback.
>
> > The author mentions that the theorem (that for a single-bidder environment, an auction with regret can be converted to an auction without regret with a revenue loss bounded by the square root of the regret) can be extended to the multi-bidder environment if using the notion of BIC. However, according to the definition, the regret is still computed in the ex-post level, which seems a bit strange. Could you elaborate more on this?
>
> Sorry for the confusion --- we were just commenting that “a variant” of Proposition 1 holds under BIC, so it’s reasonable to conjecture that it holds under DSIC as well (we were not intending to formally state this variant, though, since the precise statement is not relevant). But the reviewer is correct: the variant for BIC requires a different definition of regret (averaged over all $V_{-i}$), which would make 0 regret consistent with BIC.
>
> >Is it true that the misreport's utility is always 0 in any Nash equilibrium of the two-player game? Or this is only true for the optimal solution?
>
> At a Nash Equilibrium, the Misreporter's utility, as defined at the end of Section 3.3 is non zero, the regret of the Misreporter is zero however. This holds whenever the Misreporter is best responding, and therefore in any Nash equilibrium (not necessarily the optimal solution).
>
> Please let us know if there are other points that need some clarification.

---

### Decision · Program_Chairs · 2021-01-07
**Final Decision**

**Decision:**

Accept (Poster)

**Comment:**

There is a lot of agreement on this paper, also reflected in the ratings. There were some technical comments initially, on the approach not being IC and interpretable, missing links to other works and technical descriptions of the network and experiments. The authors cleared up many of these issues though with their responses, providing good arguments in favor of their work. In general, reviewers agree the paper would be interesting to be included in ICLR.